# Second-Generation JK-206 Targets the Oncogenic Signal Mediator RHOA in Gastric Cancer

**DOI:** 10.3390/cancers14071604

**Published:** 2022-03-22

**Authors:** Myeonghun Beak, Sungjin Park, Jin-Hee Kim, Hyo Jin Eom, Ho-Yeon Lee, Yon Hui Kim, Jinhyuk Lee, Seungyoon Nam

**Affiliations:** 1College of Medicine, Gachon University, Incheon 21565, Korea; beak98mh@gachon.ac.kr; 2Gachon Institute of Genome Medicine and Science, Department of Genome Medicine and Science, Gachon University Gil Medical Center, Gachon University College of Medicine, Incheon 21565, Korea; oscarpark@gachon.ac.kr; 3AI Convergence Center for Medical Science, Gachon University, Seongnam 13120, Korea; 4Yonsei Institute of Pharmaceutical Sciences, College of Pharmacy, Yonsei University, Incheon 21983, Korea; jinhee821@yonsei.ac.kr; 5Research and Development Department, Corestem Inc., Seongnam 13486, Korea; hyojin3149@gmail.com; 6Genome Editing Research Center, Korea Research Institute of Bioscience and Biotechnology (KRIBB), Daejeon 34141, Korea; hoyeonlee0831@gmail.com; 7Department of Bioinformatics, KRIBB School of Bioscience, Korea University of Science and Technology (UST), Daejeon 34113, Korea; 8Department of Biomedical Science, Hanyang Biomedical Research Institute, Hanyang University, Seoul 04763, Korea; yonhuisarahkim@gmail.com; 9Department of Life Sciences, Gachon University, Seongnam 13120, Korea; 10Department of Health Sciences and Technology, GAIHST, Gachon University, Incheon 21999, Korea

**Keywords:** gastric cancer, RHOA, JK-206, pharmacogenomics, Rhosin, transcriptomics

## Abstract

**Simple Summary:**

Ras homologous A (RHOA), a signal mediator and a GTPase, is associated with the progression of gastric cancer (GC). We present novel RHOA inhibitors designed for greater anti-GC potency by means of lead optimization. The RHOA → BIRC5 signaling circuit was found to be a new therapeutic strategy for regulating GC proliferation and migration.

**Abstract:**

Ras homologous A (RHOA), a signal mediator and a GTPase, is known to be associated with the progression of gastric cancer (GC), which is the fourth most common cause of death in the world. Previously, we designed pharmacologically optimized inhibitors against RHOA, including JK-136 and JK-139. Based on this previous work, we performed lead optimization and designed novel RHOA inhibitors for greater anti-GC potency. Two of these compounds, JK-206 and JK-312, could successfully inhibit the viability and migration of GC cell lines. Furthermore, using transcriptomic analysis of GC cells treated with JK-206, we revealed that the inhibition of RHOA might be associated with the inhibition of the mitogenic pathway. Therefore, JK-206 treatment for RHOA inhibition may be a new therapeutic strategy for regulating GC proliferation and migration.

## 1. Introduction

Gastric cancer (GC) is the fifth most prevalent and fourth most lethal cancer worldwide [1]. Although the first-line treatment for GC is endoscopic resection or surgery, the overall 5-year survival rate for GC is only 30% in the United States [2]. Early detection is critical for favorable outcomes and minimizing residual disease in GC; however, an endoscopic screening test for GC is not routinely performed in most Western countries. Consequently, a new and effective therapeutic option (e.g., trastuzumab [3]) for GC is emerging. Personalized medicine may be the best option for improving survival. Nevertheless, in clinical trials, most targeted therapies for GC have been disappointing [3,4,5].

Ras homologous A (RHOA), a small GTPase of the Rho family, plays crucial roles in oncogenic processes, including proliferation, migration, and invasion [6]. In addition, RHOA mediates epithelial–mesenchymal transition and is a destabilizer of microtubules [7]. Previously, we reported that RHOA is the hub mediator of GC progression and a biomarker and therapeutic target for GC [8,9]. Consequently, we designed RHOA inhibitors and evaluated their in vitro efficacy. We previously found that among JK-122–125, JK-122 showed superior suppression of cell viability [8]. In a follow-up study, we identified that JK-136 and JK-139 inhibited cell migration through the lead optimization of JK-122 [10]. Despite the potential of RHOA as a therapeutic target for GC, small-molecule RHOA inhibitors are underexplored [8,10] and the anticancer mechanism of RHOA inhibitors remain to be further elucidated.

In this study, we aimed to design novel RHOA inhibitors by means of the lead optimization of JK-136/JK-139 and elucidate the pathways involved in their anti-cancer mechanisms in GC using transcriptome-based network analysis.

## 2. Materials and Methods

### 2.1. Cell Culture

The following human GC cell lines were used within 6 months of resuscitation: AGS (ATCC, Mansfield, VA, USA), SNU216, SNU601, SNU668 (KCLB), and MKN1 (Riken, Tokyo, Japan). The cells were cultured in RPMI-1640 medium (Invitrogen, Carlsbad, CA, USA) supplemented with 10% fetal bovine serum (FBS; HyClone, Logan, UT, USA) and maintained at 37 °C and 5% CO_2_. Cell line identities were validated using short tandem repeat profiling (ATCC).

### 2.2. Synthesis of Hydrazide Derivates

All reactions that were sensitive to air or moisture were conducted under nitrogen. All reagents were purchased from Sigma-Aldrich (St. Louis, MO, USA) and Tokyo Chemical Company (Tokyo, Japan). All anhydrous solvents were distilled over CaH_2_, P_2_O_5_, or Na/benzophenone prior to the reaction, unless otherwise stated. Analytical thin-layer chromatography (TLC) was performed using commercial precoated TLC plates (Silica Gel 60, F_254_) purchased from Merck KGaA (Darmstadt, Germany). Spots were detected by viewing under UV light (254 nm) or colorizing with charring after dipping in phosphomolybdic acid in ethanol or potassium permanganate aqueous solution. Flash column chromatography was performed on Silica Gel 60 (~0.040–0.063 mm, 230–400 mesh, Merck). Infrared spectra were recorded on an Agilent Cary 670 FTIR instrument (Agilent Technologies, Santa Clara, CA, USA). ^1^H NMR spectra (DMSO-*d*_6_) were recorded on an Agilent 400-MR spectrometer (400 MHz; Agilent Technologies). The chemical shifts are reported in parts per million (δ) units relative to the solvent peak. The ^1^H NMR data are reported as peak multiplicities: s for singlet; d for doublet; dd for doublet of doublets; ddd for doublet of doublet of doublets; t for triplet; pseudo t for pseudo triplet; brs for broad singlet; and m for multiplet. The coupling constants are reported in Hertz. ^13^C NMR spectra (DMSO-*d*_6_) were recorded on an Agilent 400-MR spectrometer (100 MHz; Agilent Technologies). Mass spectra were recorded using an electrospray ionization source in methylene chloride or methanol.

The general procedure for the synthesis of hydrazides was as follows: a mixture of 4-phenoxybenzaldehyde for JK-201~214 (1 mmol), 6-(benzo[*d*][1,3]dioxol-5-yl)picolinaldehyde for JK-301–314 (1 mmol), and an appropriate hydrazide compound (1 mmol) in methanol (or ethanol) was stirred at room temperature or heated under reflux for 0.5 h–4 d. The progress of the reaction was monitored using TLC. After completion of the reaction, the contents were cooled to room temperature and poured into ice-cold water (5 mL) while stirring. The solid was filtered, dried, and purified by recrystallization using methanol (or ethanol) to obtain the hydrazide products JK-201–214 and JK-301–314. For detailed analytical data of the compounds, please refer to Method S1.

### 2.3. Cell Viability Assays

Cells were seeded at a density of 3000/well in 96-well plates and exposed to various concentrations of RHOA inhibitors and DMSO as control for 72 h. The effects of JK-201–214 and JK-301–314 on cell viability were assessed using the 3-(4,5-dimethylthiazol-2-yl)-5-(3-carboxymethoxyphenyl)-2-(4-sulfophenyl)-2H-tetrazolium (MTS; Promega, Madison, WI, USA) assay. The half-maximal inhibitory concentration (IC_50_) was defined as the dose required for 50% cell growth inhibition compared to the DMSO control. IC_50_ values represent the means of triplicate wells from three independent experiments for each drug concentration.

### 2.4. Migration Assay

AGS (1 × 10^5^), MKN1 (2 × 10^5^), SNU216 (2 × 10^5^), SNU668 (2 × 10^5^), SNU620 (2 × 10^5^), and SNU601 (2 × 10^5^) cells were seeded into Boyden chambers (0.8-μm pore size; Corning Life Sciences, Corning, NY, USA). Medium containing FBS was used as a chemoattractant in the bottom chamber. Migrated cells were stained using a Diff Quick Staining Kit (Thermo Fisher Scientific, Waltham, MA, USA) and photographed under an inverted microscope. The number of migrated cells in three microscopic fields per well was compared to that of control cells, yielding the average %migration.

### 2.5. Docking Simulations

In silico docking simulations of RHOA with JK inhibitors were performed using AutoDock Vina [11]; the three-dimensional structure of RHOA was obtained from the Protein Data Bank (accession number 1X86). The structure of RHOA consists of two chains: the Rho guanine-nucleotide exchange factor (chain A) and the RHOA structure (chain B). For the docking simulation, the RHOA structure was obtained from the complex. Two JK structures (JK-206 and -312) were drawn and minimized using the Marvin program (ChemAxon; http://www.chemaxon.com, accessed on 30 December 2021; v5.11.4, 2012). For the docking simulation, the pockets in the RHOA complex were searched using the Pck pocket detection program (http://schwarz.benjamin.free.fr/Work/Pck/home.htm, accessed on 30 December 2021). Five pockets greater than 10.0 Å^3^ in volume were found in the complex. The docking simulations were focused on these pockets, which consisted of 30 pocket residues. Two JK structures were placed in 30 pocket residues and run ten times with different random seeds; subsequently, 300 simulations were carried out. The box size with a length of 15 Å was used to prevent the inhibitors from drifting from the center of the pocket residue. With 300 docking poses, clustering based on the center of mass was performed to categorize these docking structures using Chemistry at HARvard Macromolecular Mechanics [12]. The compounds were ranked according to the following parameters: the lowest energy of the group, the largest number of the group, and the lowest energy conformation in the group.

### 2.6. Gene Expression and Differentially Expressed Gene (DEG) Analyses

Following JK-206, -312, and DMSO treatments on AGS, MKN1, and SNU601 cells, total cellular mRNA was isolated using RNeasy kits (Qiagen, Valencia, CA, USA), reverse transcribed, and hybridized to the Affymetrix GeneChip Human 2.0 ST Array (Thermo Fisher Scientific, Waltham, MA, USA) according to the manufacturer’s instructions.

We performed DEG analysis using a microarray of GC cells treated with JK-206/-312 versus DMSO. For GC cell samples treated with DMSO, we used three samples (GEO accession numbers: GSM3984792, GSM3984796, and GSM3984800) from our previous dataset (GEO accession number: GSE135068) [10].

To identify the functional contexts of DEGs, we investigated their biological pathways using Molecular Signatures Database v7.4 [13] and Gene Set Enrichment Analysis v4.1.0 [14] based on the hallmark gene set [13] for group-wise pathway analyses, and Gene Set Variation Analysis (GSVA) [15] for a sample-wise pathway analysis. In the context analysis, an adjusted *p* value (false discovery rate (FDR)) was used for statistical significance.

### 2.7. Validation of mRNA Expression Patterns of DEGs in Independent Transcriptome Datasets

To validate the mRNA expression patterns of DEGs in publicly available independent datasets, we inspected the expression levels of *RHOA* and DEGs using publicly available transcriptome datasets, GSE135068 [10] and the expression level dataset of the Cancer Cell Line Encyclopedia (CCLE) [16], GSE110237 [17], and GSE83913 [18].

### 2.8. Protein–Protein Interaction (PPI) Network Construction

To explore the interactions between our RHOA inhibitors and perturbed gene sets in GC cells treated with the JK series of compounds versus DMSO, we constructed a PPI network with the resultant DEGs identified from the gene set analysis and participant genes of the RHOA signaling pathway. The RHOA signaling pathway genes were manually selected from our previous review [19]. The PPI network was constructed using the STRING [20] and Biological General Repository for Interaction Datasets (BioGRID) databases [21]. Gene ontology analysis was performed using the STRING database to investigate the functional role of the PPI network. An adjusted *p* value (FDR) < 0.05 was considered statistically significant for this investigation. An independent GC dataset, GSE36968 [22], was used to inspect gene expression patterns among DEGs in the PPI network between GC tumor tissues (tumor group) and normal tissues adjacent to the tumor (control group) using DEG and correlation analyses. Statistical tests between the two groups were performed using the two-sided paired *t*-test. Statistical significance was set at *p* < 0.05. Correlation analysis was performed using the Pearson correlation coefficient (PCC). PCC > |0.5| and *p* < 0.05 were considered statistically significant.

### 2.9. Statistics

The DEGs satisfied the following conditions: (i) the cut-off value of fold-change (FC) was greater than 1.3 or less than 1/1.3; and (ii) statistical tests among GC cell groups were performed using the two-sided paired *t*-test, with statistical significance at *p* value < 0.05.

An adjusted *p* value (FDR) < 0.05 was considered statistically significant.

## 3. Results

### 3.1. Design and Synthesis of RHOA Inhibitors

We designed and synthesized a series of new anti-GC compounds of varying sizes, with various substituents, and with different electronic effects on phenyl-ring and heterocyclic systems for the structural modification of the benzoyl group in JK-136 and JK-139. In our previous work [8], we performed the biological evaluation of the five different compounds of an RHOA inhibitor of our own design (JK-121, -122, -123, -124, and -125) to show different chemical properties. Out of the five compounds, the compound with the benzoyl group (JK-122) exhibited the best growth inhibition in in vitro assays of GC cells and the best binding affinity to RHOA. Based on these previous results, we assumed the benzoyl group to be the necessary skeletal group of the compound and decided to introduce other functional groups on it for further optimization of our novel RHOA inhibitors. The synthetic strategy for the hydrazides JK-201–214 and JK-301–314 is displayed in Figure 1.

JK-201 and JK-301 were designed to introduce a sulfonyl group instead of a carbonyl group in the hydrazide spacers of JK-136 and JK-139. JK-202–207 and JK-302–307 possess structural diversity with different substituents in the phenyl ring of these compounds. In particular, JK-206 and JK-306 were designed to contain oxygen, forming a 1,3-dioxole ring connected to their phenyl rings. JK-208–210 and JK-308–310 were designed to form aryl heterocyclic systems. We further extended the spacer length with methylene or pyrrolidine moieties with more flexible hydrazide functional group systems, leading to the design of JK-211–214 and JK-311–314.

For the synthesis of the target hydrazides JK-201–214, commercially available 4-phenoxybenzaldehyde was treated with appropriate hydrazide reagents in methanol or ethanol at room temperature or under reflux conditions. The reaction of 6-(1,3-benzodioxol-5-yl)-2-pyridinecarbaldehyde with appropriate hydrazide reagents in methanol or ethanol at room temperature or heating conditions resulted in the creation of the hydrazide products JK-301–314.

### 3.2. Selection of RHOA Inhibitors

We evaluated the IC_50_ values of the final RHOA inhibitors (Rhosin (Millipore, Burlington, MA, USA), a nonclinical RHOA inhibitor [23], JK201–214, and JK301–314) against GC cell lines (Figure 2a and Appendix A). The designed compounds, including JK-206 and JK-312, exhibited IC_50_ values of less than 2.5 µM (Figure 2a and Appendix A). Overall, all compounds showed substantially lower cell viability compared with Rhosin, and JK-206 and JK-312 showed the best cell viability inhibition potency (Figure 2a and Appendix A). In our previous study [8], we assessed the dependency of RHOA signaling in cell lines. RHOA expression levels varied in most GC cells as AGS, MKN1, and SNU601 cells had low, moderate, and high RHOA expression, respectively. Performing a RHOA gene mutation through cancer.sanger.ac.uk (accessed on 30 December 2021), it was found that AGS has a p.E64del (amino acid) mutation on *RHOA*, but the MKN1 and SNU601 cell lines have no mutation on *RHOA*. Based on these results, we strategically selected candidate compounds based on the drug treatment results in AGS, MKN1, and SNU601 cells. We determined the efficacy of the compounds using cell density assays after treatment with 2 µM of each compound. In AGS and SNU601 cells, JK-206 exhibited the best inhibitory performance compared to other JK-200 series compounds and JK-312 showed the best inhibitory performance compared to other JK-300 series compounds (Appendix A). In migration assays, JK-206 and -312 remarkably inhibited wound healing in AGS, MKN1, SNU601, SNU216, and SNU668 cells (Figure 2b). Based on these results, we selected JK-206 and JK-312 for further experimentation.

### 3.3. Docking Simulations Indicated That JK-206 and -312 Bind with the Key Residue of RHOA

Docking simulations of RHOA with JK-206 and -312 were performed. The RHOA protein’s molecular representation is described on Figure 3a. The lowest energy conformers were found, as shown in Figure 3b,c. The docking energies of JK-206 and -312 are −6.8 and −7.0 kcal/mol, respectively, indicating that the inhibitors can tightly bind to the RHOA structure. The binding pocket in RHOA that interacts with JK-206 and -312 is the GDP-binding site on which the original ligand, GDP, of RHOA interacts. The docking of the JK compounds on GDP-binding pocket can compete against the original ligand, GDP. The competition inhibition may prevent the activation of the interaction between RHOA and LARG. The hydrogen bond patterns between the inhibitor and RHOA are shown in Figure 3d,e. A detailed analysis of the hydrogen bond is shown below the hydrogen bond patterns.

### 3.4. Gene Set Analysis Revealed That JK-206 Treatment Perturbed Myc Targets and G2/M Checkpoint

We performed gene set analyses using gene expression datasets from GC cells (AGS, MKN1, and SNU601). GC cells were treated with the selected RHOA inhibitors JK-206 and -312 (experimental groups) and DMSO (control group). Appendix A includes a list of DEGs that are common and uncommon to JK-206- and JK-312-treated (versus DMSO) GC cells. JK-206 treatment of GC cells showed that Myc targets (*p* value < 0.0001 from the gene set investigation) and G2/M checkpoint (*p* value < 0.0001 from the gene set investigation) gene sets were significantly enriched compared to the control group (Figure 4a). Of the common genes between the Myc targets and the G2/M checkpoint gene sets, 10 DEGs were identified in JK-206-treated GC cells compared to the control group, and all of them were downregulated (Figure 4b). The expression levels of *RHOA* were also decreased significantly (*p* < 0.05) after the inhibition of RHOA by JK-206 treatment compared to DMSO treatment in GC cells (Figure 4c). The genes relating to Myc targets (i.e., *BIRC5*, *H2AFZ*, *HIST1H2BK*, *KIF15*, *UBE2S*, and *FBXO5*; Appendix A) and G2/M checkpoint (i.e., *RUVBL2*, *H2AFZ*, *EIF1AX*, *SNRPD3*, and *SET*; Appendix A) were down-regulated in JK-312-treated GC cells compared to DMSO-treated GC cells. However, the gene set analysis failed in JK-312-treated GC cells compared to the control group and we could not obtain statistically significant gene sets. We additionally performed a sample-specific pathway-activity analysis for the expression data of JK-312-treated cells, instead of using the GSEA (the group-wise pathway enrichment analysis). In the results of the sample-wise pathway analysis conducted using the gene set variation analysis (GSVA) [15], it was found that JK-312-treated GC cell lines were also suppressed in hallmark gene sets of Myc targets, G2/M checkpoint, and E2F targets such as JK-206-treated GC cell lines compared to DMSO-treated GC cell lines (Appendix A).

### 3.5. Validation of mRNA Expression Patterns of DEGs Related to Myc Targets and G2/M Checkpoint in Independent Datasets

To determine whether Myc-target genes and G2/M-checkpoint related genes were suppressed due to the RHOA inhibition, we inspected publicly available independent datasets. For validating the expression levels of *RHOA*, Myc-target genes, and G2/M-checkpoint-related genes, we re-visited and inspected the transcriptome data (GEO accession: GSE135068) [10] from our previous study of other RHOA inhibitors (JK-136 and -139). We found that, upon RHOA inhibition by JK-136 and -139, most of the expression levels of *RHOA* and DEGs related to Myc targets and G2/M checkpoint (depicted in Figure 4b) were consistently decreased (Appendix A).

For another validation, since the RHOA inhibitors (JK-206, -312, -136, and -139) demonstrated down-regulation of *RHOA* and the DEGs (depicted in Figure 4b), we assumed that there were positive correlations between *RHOA* and the DEGs. In 13 GC cell lines from CCLE [16], we calculated Pearson’s correlation, and found that most of the correlations were positive (Appendix A). We also inspected other publicly available transcriptomic datasets for *RHOA* inhibition by siRNA and knockout in gastric and prostate cancer cell lines (GEO accessions GSE110237 and GSE83913, respectively) [17,18]. Dataset GSE110237 was derived from experiments of *RHOA* knockdown in GC cells [17] and dataset GSE83913 was derived from experiments of *RHOA* knockout in prostate cancer cells [18]. We confirmed that most of the DEGs were also down-regulated after knockdown of *RHOA* in GC cell lines and after knockout of *RHOA* in prostate cancer cell lines (Appendix A).

### 3.6. Network Construction Revealed That JK-206 Perturbed Microtubule Formation and the Cell Cycle

In this study, since the group-wise and sample-specific pathway analyzes showed consistencies in GC cell lines treated with JK-206, we performed further investigation only on JK-206-treated GC cells compared to the control group. JK-206 was designed and optimized to inhibit RHOA expression. To explore the biological mechanism perturbed by JK-206 treatment in GC cells, we constructed PPI networks using the 10 genes (Figure 4b) and RHOA signaling pathway genes as inputs in the STRING [20] and BioGRID [21] databases. In the PPI network (Figure 5a), the RHOA signaling pathway and the 10 DEGs (Figure 4b) out of the common genes between the two gene sets (Myc targets and G2/M checkpoint) were found to interact with each other through RAC1 and CDK4. Most of the gene expression levels in the RHOA signaling pathway were also downregulated (Figure 5a).

The functional contexts of JK-206 treatment in GC cells were found to involve WNT signaling, Rho GTPases, cell motility, and the cell cycle (Figure 5b). The gene expression levels of microtubules (i.e., baculoviral IAP repeat-containing 5 (*BIRC5*), F-Box protein 5 (*FBXO5*), RuvB-like AAA+ ATPase 2 (*RUVBL2*), and kinesin family member 15 (*KIF15*)) were downregulated in JK-206-treated GC cells compared to the control group (Figure 4b and Figure 5b). Therefore, JK-206 treatment had an inhibitory effect on microtubule activity in GC cells, which is in agreement with our migration assay results (Figure 2b). Similarly, our previously reported RHOA inhibitors (i.e., JK-136 and -139) regulate actin polymerization and cell migration [10]. Similar to the functional contexts identified in this study, cell-cycle-related gene sets (i.e., M phase and nuclear division) were identified in the functional context analysis in our previous study [10]. These gene sets comprised the protein products H2A.Z variant histone 1 (H2AFZ), HIST1H2BK, SET nuclear proto-oncogene (SET), UBE2S, FBXO5, BIRC5, and RUVBL2 of downregulated DEGs (Figure 4b and Figure 5b). In line with these results, JK-206 treatment exerted an inhibitory effect on cell-cycle-related activity against GC cells.

To determine whether the expression levels of the genes identified in the PPI network were replicated in a GC patient dataset, we performed DEG and correlation analyses with the 23 genes in Figure 5a using an independent GC patient dataset (GEO accession number: GSE36968) [22]. We assumed that the downregulated genes in the PPI networks (Figure 5a) were upregulated in the tumor versus control groups in the dataset GSE36968. In fact, *BIRC5* was significantly upregulated in the tumor group compared with the control group (FC = 2.45, *p* value < 0.01; Figure 5c) and its expression level showed a significantly positive correlation with that of *RHOA* (r = 0.56, *p* value = 0.001; Figure 5c). Furthermore, the expression levels of *RAC1, CDK2, RUVBL2*, and *MET* (shown in Figure 5a) were upregulated in the tumor group compared to the control group and showed a noticeably positive correlation with the expression level of *RHOA* (Appendix A). Considering the positive expression correlation of *BIRC5*, *RAC1, CDK2, RUVBL2*, and *MET* with *RHOA* in the GC patient dataset, the downregulation of the genes with JK-206 treatment in the GC cell models indicates that RHOA signaling may be involved in upstream regulation of genes in GC.

## 4. Discussion

In this study, we designed and synthesized a series of new RHOA inhibitors to evaluate and improve their biological activity [10]. Although the RHOA signaling pathway is considered a potential therapeutic target for GC, the effects of RHOA inhibitors on cancer cells are yet to be elucidated. Among the new compounds we studied, JK-206 could effectively inhibit cancer cell survival and migration. To determine the mechanism of action of JK-206, we analyzed transcriptomic data of GC cell lines treated with JK-206 compared to those treated with DMSO. We found that JK-206 suppressed biological contexts associated with WNT, Rho GTPases, Myc targets, G2/M checkpoint, and microtubule-related gene sets. In our previous study, we confirmed that the RHOA signaling pathway plays a crucial role in GC progression [9]. RHOA interacts with c-Myc, resulting in synergistic reinforcement of the expression of cancer phenotypes and behaviors favoring cancer development, including cell migration, invasion, and metastasis [24,25].

Through DEG and gene set analyses, 10 DEGs were found to be associated with two gene sets related to cell motility and cycle (i.e., Myc targets and G2/M checkpoint) in GC cells treated with JK-206 compared to those treated with DMSO (Figure 4b). These genes are involved in apoptosis inhibition (*BIRC5*) [26], oncogene activation (*H2AFZ* and *SET*) [27,28], protein degradation regulation (*FBXO5* and *UBE2S*) [29,30], and carcinogenesis (*RUVBL2* and eukaryotic translation initiation factor 1A X-linked (*EIF1AX*)) [31,32]. In our networks, the inhibition of RHOA signaling by JK-206 treatment may induce the suppression of Myc targets through downregulation of *BIRC5*, *H2AFZ*, *HIST1H2BK*, *KIF15*, *UBE2S*, and *FBXO5* in GC cells. KIF15 promotes cell mitosis and structural assembly [33,34] and its high gene expression is associated with poor prognosis in GC patients; on the contrary, the suppression of its expression inhibits GC cell progression, promotion of apoptosis, and cell cycle arrest [35]. In multiple GC dataset analyses, *H2AFZ* was identified as a crucial hub gene in the GC co-expression network, which indicated that *H2AFZ* has high degrees of interaction with multiple participant genes and may function as a control node in the GC co-expression network [36]. BIRC5, also known as survivin, is an apoptosis inhibitor [37] and acts as a resistance factor to anticancer therapies [38,39]. BIRC5 is a molecular marker of the poor prognosis in lung, pancreatic, and breast cancers [40]. It is known to increase lymph node metastasis [41] and is associated with poor clinical outcomes in GC [42]. Increased expression of BIRC5 is a risk factor for cancer progression and poor outcomes in breast cancer [43]. Though BIRC5 has been validated as a target of cancer drugs [44], the number of BIRC5 inhibitors available for clinical testing is limited [45]. As BIRC5 participates in large cellular networks, pathway inhibitors [46] have been proposed as promising alternatives to a single protein inhibitor [45]. In this study, *BIRC5* was remarkably suppressed in GC cells when RHOA was inhibited by the RHOA inhibitor JK-206 (Figure 4b). BIRC5 was found to interact with the RHOA signaling pathway (Figure 5a) and was associated with mitotic activity (Figure 5b). In the independent GC dataset, *BIRC5* was highly expressed in the tumor group compared to the control group, and its expression level was substantially correlated with that of *RHOA* (Figure 5c). Therefore, JK-206 could inhibit RHOA and, thus, suppress BIRC5.

Further investigation of *BIRC5*, *H2AFZ*, *HIST1H2BK*, *KIF15*, *UBE2S*, and *FBXO5* is crucial to broaden the understanding of cancer invasion and migration, and to promote the discovery and evaluation of new therapeutic targets.

RHOA activity increases in pre-anaphase mitotic cells and participates in G2/M transition through mitotic cell rounding and de-adhesion [47]. The inhibition of RHOA leads to the inhibition of G2/M transition via the interference of mitotic cell rounding and de-adhesion [47,48]. In this study, JK-206 treatment dysregulated the G2/M checkpoint in GC cells. Disturbance of the G2/M checkpoint induces G2/M arrest, genomic instability, and subsequently, cell apoptosis [49]. JK-206 treatment may drive G2/M checkpoint arrest or catastrophic mitosis, resulting in subsequent apoptosis, which was confirmed by a notable decrease in cell viability in the cell viability assay (Figure 2a and Appendix A). Modifications of apoptosis- and cell-cycle-related pathways were also derived using our previously designed RHOA inhibitors (JK-136 and -139) [10]. Cross-talk between the RHOA signaling pathway and G2/M checkpoint is yet to be elucidated. Among the participant genes of the G2/M checkpoint gene set, *RUVBL2*, *EIF1AX*, and *SET* are known to play important roles in the cross-talk between RHOA signaling and cell cycle activity [50,51,52]. RUVBL2 is associated with poor prognosis in multiple types of cancer [52,53,54]. RUVBL2 is involved in DNA replication and its inhibition results in cancer cell death via cell cycle (i.e., S-phase) arrest and subsequent replication catastrophe in non-small cell lung cancer [52]. EIF1AX stimulates cell proliferation by inducing cell cycle transition (i.e., G1/S) via the inhibition of p21 expression, resulting in poor prognosis in breast cancer patients [51]. SET regulates the G2/M transition by modulating CDK1 [50], and it results in tumor progression and poor clinical outcomes in GC [55]. Therefore, SET is considered a potential therapeutic target [50,56,57]. In future studies, the specific roles of identified DEGs in the G2/M checkpoint gene set (i.e., *RUVBL2*, *H2AFZ*, *EIF1AX*, *SNRPD3*, and *SET*) need to be investigated to understand apoptosis or mitotic catastrophe induced by G2/M checkpoint alterations and discover new therapeutic targets.

The migration of cancer cells requires the reformation of microtubules, and RHOA is essential for cell motility, adhesion, and the regulation of actin and microtubules [58]. In our PPI network construction, JK-206 was found to be associated with the microtubule formation and function in GC cells, which was also confirmed by the remarkable reduction in cell migration in the cell migration assay (Figure 2b). We hypothesize that the downregulation of *BIRC5*, *FBXO5*, *KIF15*, and *RUVBL2* led to the perturbation of microtubule dynamics with JK-206 treatment in GC cells. Considering the biological roles of microtubules in the cell cycle, JK-206-induced perturbation of microtubule dynamics may have affected cell cycle transition, especially at the G2/M checkpoint. Inhibitors also serve as chemical probes to reveal biological mechanisms [59]. JK-206 showed good efficacy in the in vitro viability assay (Figure 2a,b), and through network analysis, we showed an association between RHOA and BIRC5 in the transcriptome dataset of JK-206-treated GC cell lines. We only performed in vitro and in silico assays to measure the efficacy of JK-206; however, the discrepancy between in vitro and in vivo experiments might be observed due to GC heterogeneity, as we previously reported [10]. Further studies based on diverse GC cell lines and xenograft models are needed.

There are limitations. In this study, the mRNA and protein expressions of RHOA and the DEGs by JK-206 were not investigated through RT-PCR and western blotting. Thus, our conclusions should be carefully interpreted.

In this study, a new RHOA inhibitor, JK-206, substantially decreased both cell viability and migration in GC cells (Figure 2a,b). Through systematic analysis of the gene expression dataset, we identified that JK-206 treatment led to suppression of the expression levels of oncogenic genes, which play key roles in apoptosis, carcinogenesis, and protein degradation. JK-206 treatment affected pathways related to the G2/M checkpoint, Myc target, and microtubule dynamics. Thus, the impact of JK-206 on multiple oncogenic pathways represents a promising therapeutic potential for GC as a pathway inhibitor.

## 5. Conclusions

In conclusion, we identified the optimized compound JK-206 as an RHOA inhibitor in GC. This study not only supports our previous extensive studies regarding RHOA as a therapeutic target in GC [8,10,60], but also suggests RHOA-mediated mitogenic pathway regulation as a new therapeutic strategy for GC.

## Figures and Tables

**Figure 1 cancers-14-01604-f001:**
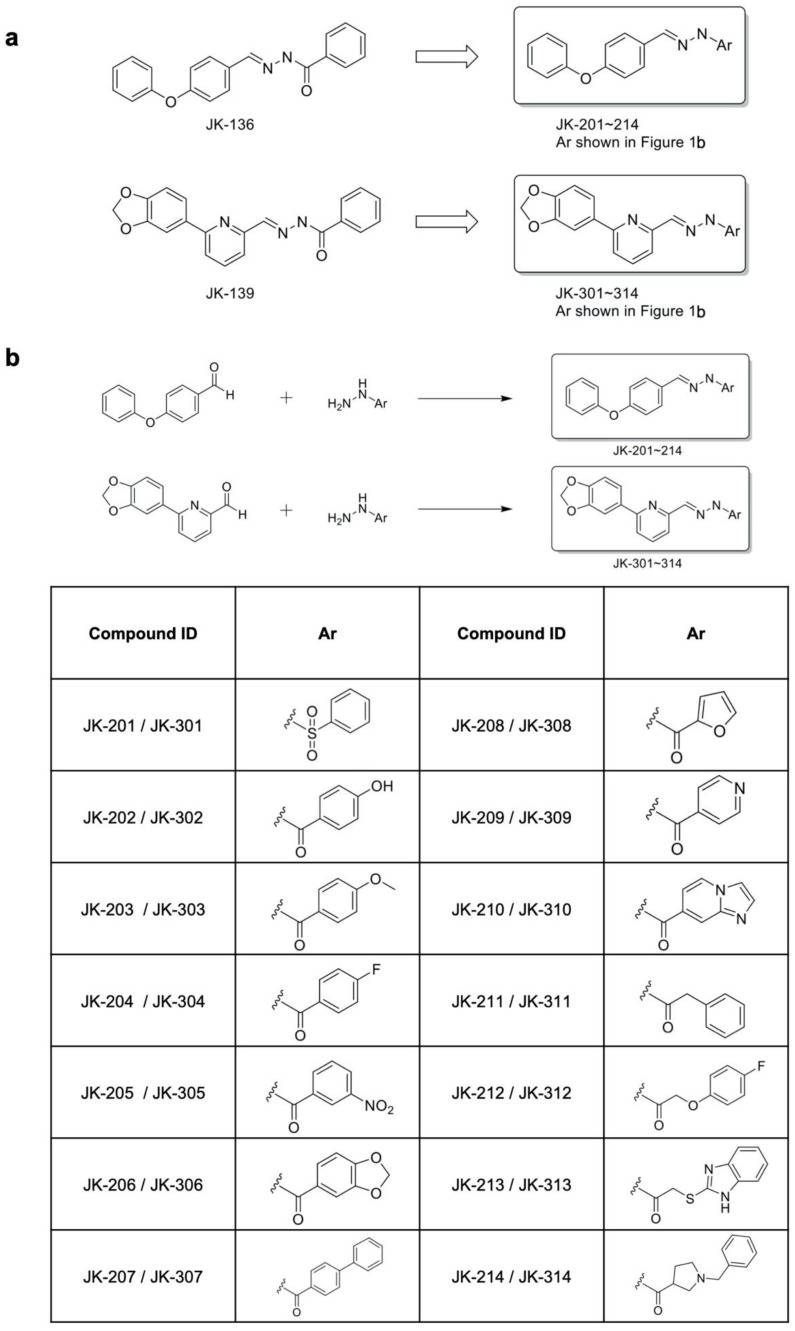
Design and synthesis of Ras homologous A (RHOA) inhibitors. (**a**) Rational design of novel RHOA inhibitors. (**b**) Synthesis of hydrazide derivatives. Reagents and conditions: methanol or ethanol, room temperature or reflux, 0.5 h–4 d.

**Figure 2 cancers-14-01604-f002:**
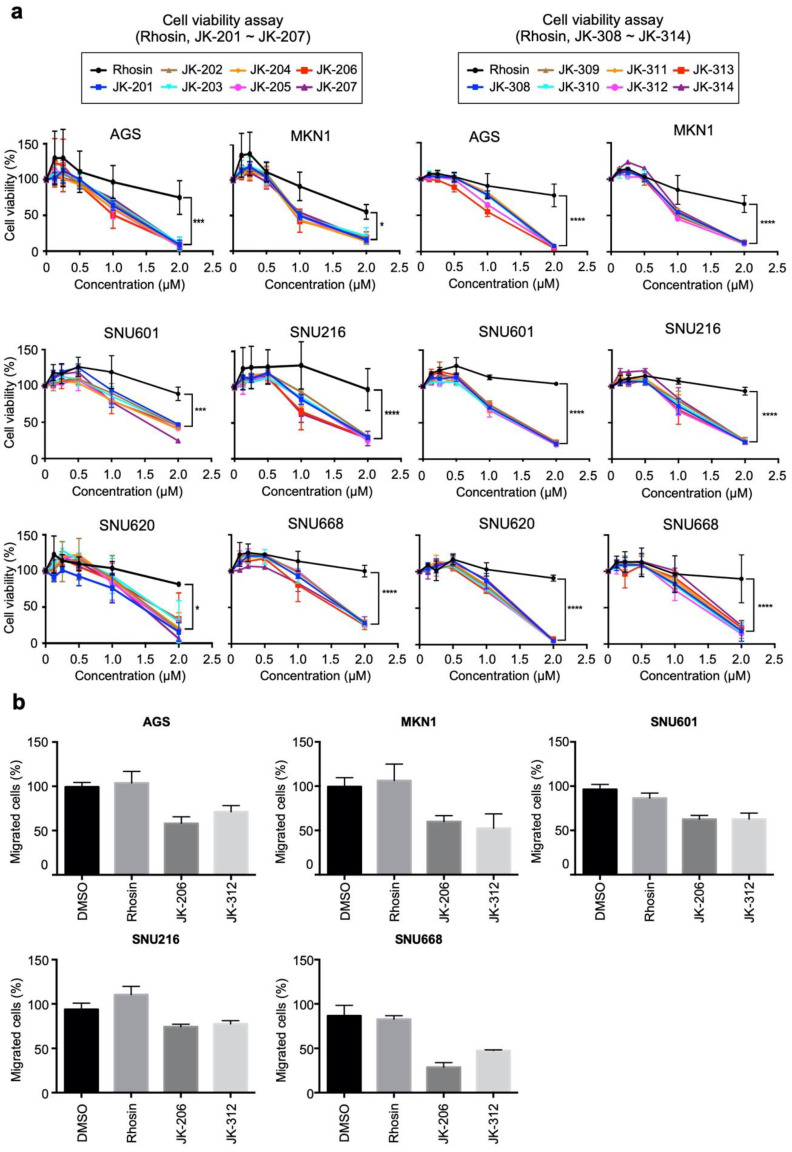
RHOA inhibitors suppressed cell growth and migration in gastric cancer (GC) cells. (**a**) The GC cell lines AGS, MKN1, SNU601, SNU216, SNU620, and SNU668 were treated with 15 small-molecule candidates including Rhosin (see also Appendix A). (**b**) The migration assay showed significant inhibition of wound healing with treatment with both JK-206 and JK-312 (* *p* value < 0.05, *** *p* value < 0.001, **** *p* value < 0.0001).

**Figure 3 cancers-14-01604-f003:**
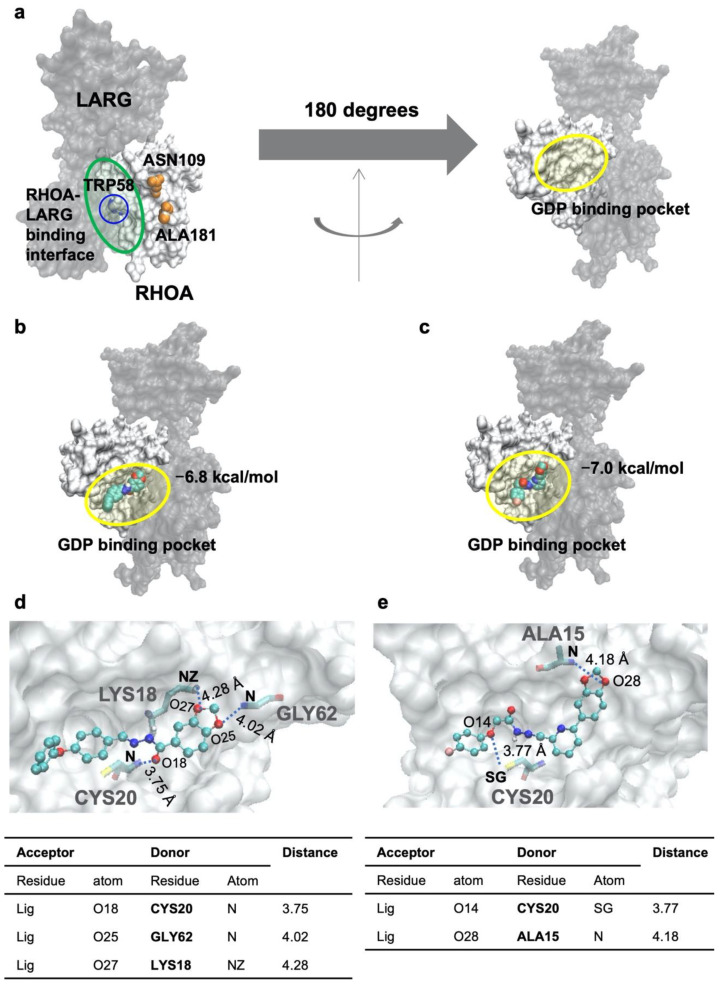
Molecular docking simulations and hydrogen bond patterns. (**a**) Molecular representation of the RHOA–LARG complex. The RHOA structure is illustrated using a white surface model. The interacting partner protein, LARG (Rho guanine nucleotide exchange factor 12), is shaded based on a black transparency surface model. The two structures are interacting on the RHOA–LARG-binding interface circled by transparent green in the left figure. The TRP58 is located on the RHOA–LARG-binding surface. The channel is formed between ASN109 and ALA181, which are illustrated by orange spheres. The GDP-binding pocket (transparent yellow circle in the right figure) exists on the opposite side (180 degrees’ rotation of the principal axis of the proteins). The docking poses (**b**) JK-206 and (**c**) JK-312 are drawn on the RHOA. The inhibitors are also drawn using space-filling models colored by atom types (oxygen: red; nitrogen: blue; carbon: cyan). The hydrogen-bond-interacting patterns of (**d**) JK-206 and (**e**) JK-312 are drawn. The inhibitors are drawn using ball-and-stick models colored by atom types. The interacting residues are drawn using a stick model colored by atom types. The interacting hydrogen bonds are drawn using dot lines with the bonding distances, with the Angstrom as the unit. Detailed information on the hydrogen bonds is tabulated below the hydrogen-bond-interacting pattern.

**Figure 4 cancers-14-01604-f004:**
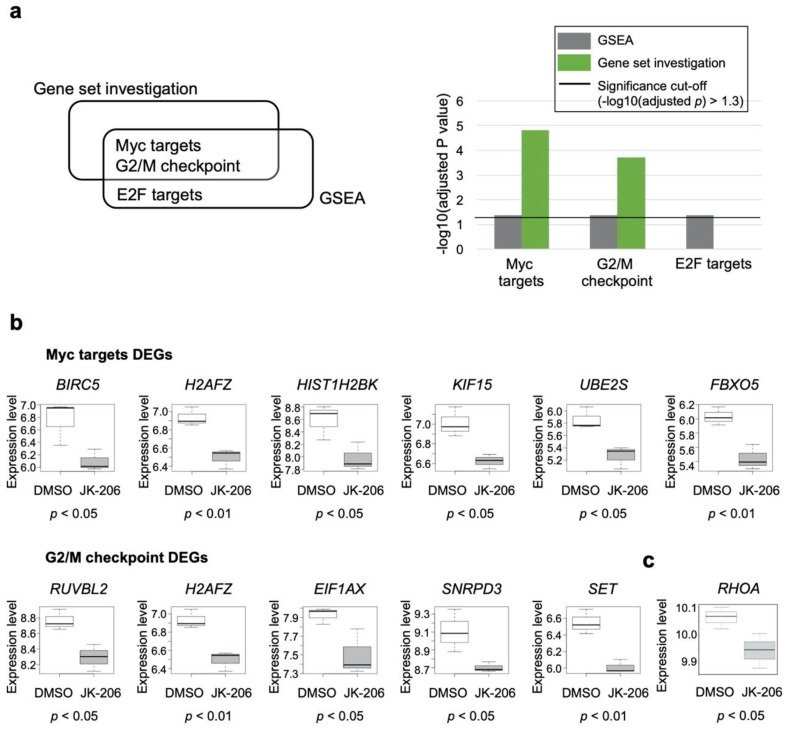
Gene set analysis of GC cells treated with JK-206 and -312 compared to those treated with dimethyl sulfoxide (DMSO). (**a**) Significantly detected gene sets of gene set analyses. (**b**) Expression profiles of the differentially expressed genes (DEGs) in the two gene sets (Myc targets and G2/M checkpoint). (**c**) Expression pattern of *RHOA* in JK-206- vs. DMSO-treated GC cells.

**Figure 5 cancers-14-01604-f005:**
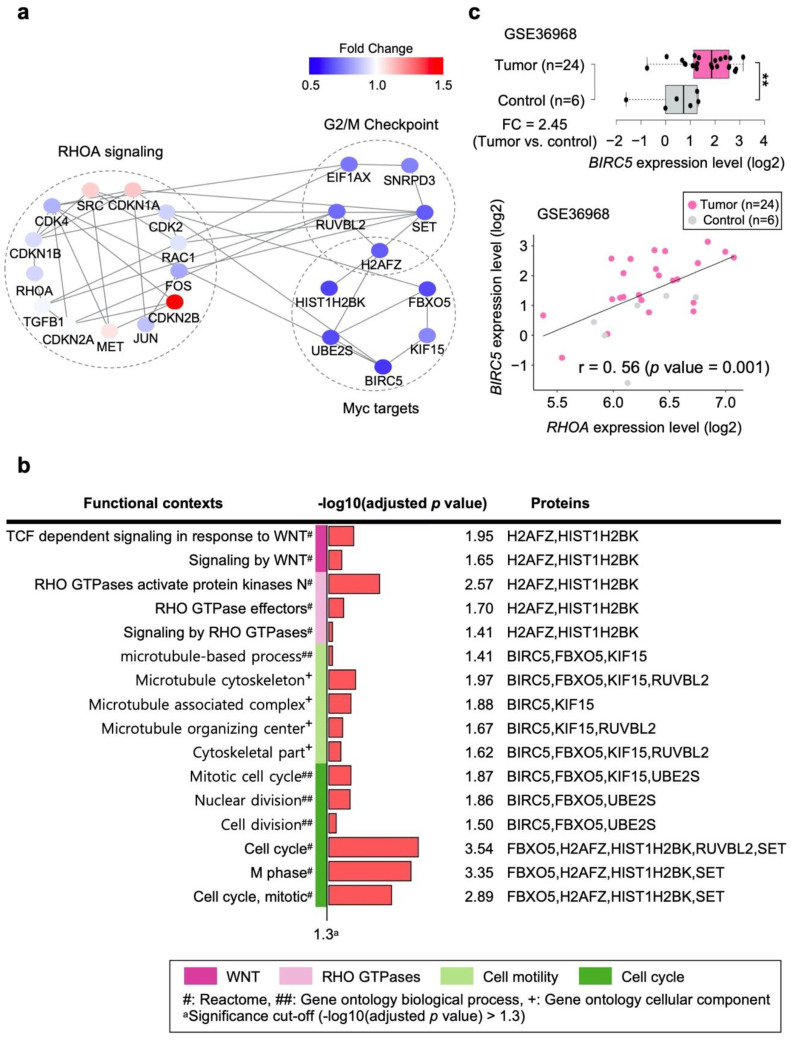
Protein-protein interaction (PPI) network construction and functional context analysis of the network of GC cells treated with JK-206 compared to those treated with DMSO. (**a**) PPI networks constructed from RHOA signaling and DEGs of GC cells treated with JK-206 compared to those treated with DMSO. (**b**) Functional contexts of GC cells treated with JK-206 compared to those treated with DMSO in the PPI network. (**c**) Differentially expressed *BIRC5* between tumor and control groups and correlation between *BIRC5* and *RHOA* in the independent GC patient dataset (GEO accession number: GSE36968). r: Pearson’s correlation coefficient. ** *p* value < 0.01.

## Data Availability

All data are publicly available. The microarray dataset in this study was deposited in the NCBI GEO repository (GEO accession GSE188908) (https://www.ncbi.nlm.nih.gov/geo/query/acc.cgi?acc=GSE188908, accessed on 30 December 2021). For GC-cell-treated DMSO, three samples (GSM3984792, GSM3984796, and GSM3984800) were downloaded from our previously deposited dataset, GSE135068 (https://www.ncbi.nlm.nih.gov/geo/query/acc.cgi?acc=GSE135068, accessed on 30 December 2021).

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
