# Peer review of "Second-Generation JK-206 Targets the Oncogenic Signal Mediator RHOA in Gastric Cancer"

_cancers, 2022, doi:10.3390/cancers14071604_

Round 1

Reviewer 1 Report

Thank the authors for making the effort to address my concerns. The sentence has been revised. I have no further comments.

Author Response

Thank the reviewer for the comment.

Reviewer 2 Report

My critiques were not addressed.

Author Response

This manuscript is a resubmission of an earlier submission. The following is a list of the peer review reports and author responses from that submission.

Round 1

Reviewer 1 Report

In this manuscript, the authors well described the pharmacological aspects of their novel and original RHOA inhibitors, JK-206 and JK-312, and demonstrated that they would be promising for the treatment of gastric cancer. Nonetheless, I would like the authors to address the following concerns.

1. Theoretical explanation is necessary for why the authors chose the benzoyl group to introduce other functional groups.

2. What is Rhosin?

3. The authors should check mutation status of the RHOA gene in the GC cell lines used in the study at the following database. https://cancer.sanger.ac.uk/cell_lines

4. Based on the docking study, the authors should discuss how RHOA function is inhibited by the compounds, and whether the compounds are capable of inhibiting mutant ROHA protein.

5. The expression levels of the genes (Figure 4b) in JK-312-treated cells should be shown.

6. Although the authors say that the GSEA were failed in JK-312-treated GC cells, DEGs can be extracted without GSEA. The authors should provide a list of DEGs common and uncommon to JK-206 and JK-312.

7. The reviewer thinks that it is possible to perform the pathway analysis for DEGs in JK-312-treated cells. The authors should perform this for JK-312 without GSEA.

8. In order to prove that the DEGs are attributable to RHOA inhibition, the authors should determine the expression of the DEGs in RHOA knockdown cells.

9. It is necessary to prove that the DEGs, at least BIRC5, are involved in the mechanism of action of JK-206 and JK-312, by knockdown or overexpression experiments.

Author Response

Please see authors' responses of Reviewer 1 in the attachment. 

Reviewer 2 Report

The authors must show the protein and mRNA expression of the differentially expressed genes mentioned in the manuscript through RT PCR and western blot. This would confirm the conclusion the authors are trying to make "pharmacological inhibition of RHOA was associated with inhibition of the mitogenic pathway, which includes BIRC5"

In the manuscript " Second-generation JK-206 targets the oncogenic signal mediator RHOA in gastric cancer" , the authors have shown the inhibitory action of JK-206 on RHOA through BIRC5. RHOA inhibition in gastric cancer is not a new topic. Therefore, authors must support their findings strongly by showing that JK-206 indeed suppresses RHOA. Therefore, the authors must show that JK-206 suppresses the protein and mRNA expression of RHOA, Myc, Myc targets (such as BIRC5, H2AFZ, HIST1H2BK, KIF15, UBE2S, and FBXO5) and G/M checkpoints through western blotting and Real-Time PCR. 

Author Response

Please see authors' responses for Reviewer 2 in the attachment.
